Intensive hunting changes human-wildlife relationships

Parsons Arielle Waldstein aparsons@lpzoo.org 1 2
Wikelski Martin 3 4 5
Keeves von Wolff Brigitta 3
Dodel Jan 6
Kays Roland 1 7
1 North Carolina State University , Raleigh , NC , United States of America
2 Lincoln Park Zoo , Chicago , IL , United States of America
3 Department of Migration, Max Planck Institute of Animal Behavior , Radolfzell , Germany
4 Centre for the Advanced Study of Collective Behaviour, University of Konstanz , Konstanz , Germany
5 Max Planck-Yale Center for Biodiversity Movement and Global Change, Yale University , New Haven , CT , United States of America
6 Konstanz , Germany
7 North Carolina Museum of Natural Sciences , Raleigh , NC , United States of America
Gandini Patricia
Electronic publication date: 2022 Oct 11
Publication date: 2022
Volume: 10
Electronic Location ID: e14159
Received 2022 Jun 20; Accepted 2022 Sep 8
Copyright: ©2022 Parsons et al.
Copyright year: 2022
Copyright holder: Parsons et al.
License: This is an open access article distributed under the terms of the Creative Commons Attribution License, which permits unrestricted use, distribution, reproduction and adaptation in any medium and for any purpose provided that it is properly attributed. For attribution, the original author(s), title, publication source (PeerJ) and either DOI or URL of the article must be cited.
License URL: https://creativecommons.org/licenses/by/4.0/

Keywords: Antipredator behavior, Ecological impacts, Europe, Hunting pressure, Landscape of fear, North America, Relative abundance, Risk allocation

Funding: The Deutsche Forschungsgemeinschaft (DFG, German Research Foundation) under Germany’s ExcellenceStrategy –EXC 2117 –422037984 The US National Science Foundation #1232442 #1319293 The US Forest Service and the North Carolina Museum of Natural Sciences This work was supported by the Deutsche Forschungsgemeinschaft (DFG, German Research Foundation) under Germany’s ExcellenceStrategy–EXC 2117–422037984, the US National Science Foundation (grant #1232442 and #1319293), the US Forest Service and the North Carolina Museum of Natural Sciences. The funders had no role in study design, data collection and analysis, decision to publish, or preparation of the manuscript.

==============================
Wildlife alter their behaviors in a trade-off between consuming food and fear of becoming food themselves. The risk allocation hypothesis posits that variation in the scale, intensity and longevity of predation threats can influence the magnitude of antipredator behavioral responses. Hunting by humans represents a threat thought to be perceived by wildlife similar to how they perceive a top predator, although hunting intensity and duration varys widely around the world. Here we evaluate the effects of hunting pressure on wildlife by comparing how two communities of mammals under different management schemes differ in their relative abundance and response to humans. Using camera traps to survey wildlife across disturbance levels (yards, farms, forests) in similar landscapes in southern Germany and southeastern USA, we tested the prediction of the risk allocation hypothesis: that the higher intensity and longevity of hunting in Germany (year round vs 3 months, 4x higher harvest/km2/year) would reduce relative abundance of hunted species and result in a larger fear-based response to humans (i.e., more spatial and temporal avoidance). We further evaluated how changes in animal abundance and behavior would result in potential changes to ecological impacts (i.e., herbivory and predation). We found that hunted species were relatively less abundant in Germany and less associated with humans on the landscape (i.e., yards and urban areas), but did not avoid humans temporally in hunted areas while hunted species in the USA showed the opposite pattern. These results are consistent with the risk allocation hypothesis where we would expect more spatial avoidance in response to threats of longer duration (i.e., year-round hunting in Germany vs. 3-month duration in USA) and less spatial avoidance but more temporal avoidance for threats of shorter duration. The expected ecological impacts of mammals in all three habitats were quite different between countries, most strikingly due to the decreases in the relative abundance of hunted species in Germany, particularly deer, with no proportional increase in unhunted species, resulting in American yards facing the potential for 25x more herbivory than German yards. Our results suggest that the duration and intensity of managed hunting can have strong and predictable effects on animal abundance and behavior, with the potential for corresponding changes in the ecological impacts of wildlife. Hunting can be an effective tool for reducing wildlife conflict due to overabundance but may require more intensive harvest than is seen in much of North America.

Introduction

All prey species must balance the trade-off between consuming resources and becoming resources for their predators such that the mere risk of predation can shape how prey behave (Miller & Schmitz, 2019). These direct and indirect effects of predation are collectively termed the “landscape of fear”, an inherently spatial concept describing elements on the landscape that an animal may perceive as high risk (i.e., predation) relative to reward (i.e., food; Laundré, Hernández & Ripple, 2010). However, fear is a simultaneous spatial and temporal concept where animals may change their antipredator behaviors relative to the spatial and temporal scale and intensity of threats, a hypothesis termed the “risk allocation hypothesis” (Lima & Bednekoff, 1999). According to this hypothesis, an animal will increase antipredator behavior, thereby reducing foraging, proportionally to the severity of the perceived threat. Taking both the spatial and temporal concepts of fear-based responses together, we can learn about how species perceive risk by comparing spatial movement and temporal activity patterns with different potential risk factors (Dröge et al., 2017). For example, elk (Cervus elaphus) in Yellowstone National Park, USA spatially avoid habitats with the highest wolf (Canis lupus) predation risk, including high-quality habitats (Creel et al., 2005). However, where predation risk is lower, elk continue to use high-quality high-risk habitats, but do so when wolves are least active during the day (Kohl et al., 2018). Thus, the form of threat, scale, predictability and longevity are all factors that shape how wildlife species respond to risks both temporally and spatially.

Humans represent a threat to wildlife which wildlife may perceived in the same way they perceive a top predator (Cromsigt et al., 2013). Numerous studies have shown wildlife altering their behaviors in response to human activities by changing their vigilance (Ciuti et al., 2012), movement rates (Proffitt et al., 2009), flight responses (Chassagneux et al., 2019) and activity patterns (Parsons et al., 2016). However, not all humans are wildlife predators, and some animal populations have habituated to lose their fear of humans (Wheat & Wilmers, 2016). Given that prey species can rapidly change antipredator behaviors in response to changes in risk (Relyea, 2003), we predict that variation in how humans hunt wildlife should affect the behavior and space use of those species.

The nature of human hunting varies greatly by region and species with regulations affecting the level of threat (firearms vs. archery or trapping), location (hunting grounds), seasonality, longevity and intensity (bag limits; the number of individual animals a hunter can harvest) of hunting pressure. For example, in Central Europe (hereafter “Europe”), hunting occurs year-round for most species, with hunting grounds being privately-owned lands, managed locally (Bubenik, 1989). By contrast, hunting in the United States and Canada (hereafter “North America”) is restricted to shorter seasons with hunting grounds being a combination of public lands managed at the state or provincial levels and private lands (McShea, 2012). Bag limits also differ between the two systems with North America limiting bags by individual hunter and Europe limiting bags at the state level by species or by property, with many properties imposing no limits, resulting in typically higher bags per unit area (Adams & Hamilton, 2011). The result is a more sustained and intensive hunting pressure in most of Europe compared to a more temporally and spatially heterogeneous (i.e., the majority of hunting occurs in large public lands with less on private lands) hunting effort in North America. Exactly how these differences in hunting regulations and pressures affect the abundance and behavior of wildlife populations is poorly understood but has implications for ecosystem health and human-wildlife interactions. Indeed, although using hunting regulations to shape animal behavior has been suggested for wildlife managment (Cromsigt et al., 2013), few studies have compared the effect of different hunting practices on wildlife behavior (e.g., Little et al., 2016).

Here we use camera traps run in two areas with similar ecologies, but different hunting regimes, to evaulate how hunting affects the degree to which animals fear humans. Specifically, we use spatio-temporal data on wildlife distribution across a range of human development (residential yards, agricultural areas and forests) to test predictions of the landscape of fear and risk allocation hypothesis. Finally, to evaulate the ecological consequences of these changes, we estimate the ecological roles of mammals at both sites. To improve our inference that any differences we observe are associated with hunting, we chose two landscapes similar in climate and land cover proportions but with different hunting systems: Baden-Württemberg (BW) in southern Germany and North Carolina (NC) in southeast USA. The spatial intensity of hunting is much higher in BW with 4.71 European roe deer (Capreolus capreolus; hereafter “roe deer”) harvested/km2 compared to 0.78 white-tailed deer (Odocoileus virginianus) harvested/km2 in NC in 2018. Furthermore, hunting in NC occurs only during short seasons (e.g., 3 months for deer, Table 1) but is longer-lasting in BW (year round; Table 1). Based on these differences, we make three predictions: (1) abundance: the more intensive hunting pressure in Germany will result in hunted species being relatively less abundant than in the USA, (2) spatio-temporal risk allocation: the more intense and sustained hunting pressure of Germany will necessitate a larger fear-based response to humans (i.e., high spatial avoidance) whereas the low-intensity, short-term hunting pressure in America will allow wildlife to maintain space with humans, instead avoiding them temporally where necessary. If wildlife species respond to hunting pressure in terms of changes in abundance, then there is potential for herbivory and/or predation rates to also change unless unhunted species with the same diets are increasing in abundance. Therefore, we make a third prediction: (3) ecological impacts: unhunted species will not compensate for lower relative abundance and lower use of human-dominated habitats for hunted species in Germany, resulting in a reduced potential for herbivory and/or predation compared to the USA.

Table 1 Hunted species in North Carolina (NC), USA and Baden-Württemberg (BW), Germany with associated bag limits, seasons lengths and annual bag for the region.

Data for NC and BW are taken from the North Carolina Wildlife Resources Commission1 and the Jagdbericht Baden-Württemberg für das Jagdjahr 2018/20192, respectively. Bags are calibrated by average body mass to show the kg hunted (in 2018) for each species. Germany has requirements for minimum and maximum numbers of hunted animals, but no bag limit per hunter.

Species	Country	Season bag limit	Season length (months)	2018 statewide harvest	Body size (kg)	kg hunted	
Heavily managed and hunted	
White-tailed deer	NC, USA	6	3	178,5541	68	12,141,668	
Bear	NC, USA	1	1	3,4761	181	629,156	
Turkey	NC, USA	3	1	26,4231	9	237,806	
European roe deer	BW, Germany	None	9	168,4012	27	4,546,827	
Boar	BW, Germany	None	Year round	47,8642*	70	3,350,480	
Hunted but not heavily managed (i.e., no bag limits imposed, longer seasons)	
Raccoon	NC, USA	None	4	65,353	6.8	444,400	
E. gray squirrel	NC, USA	None	4	219,207	1.8	394,573	
Coyote	NC, USA	None	Year round	31,808	12	381,700	
E. cottontail	NC, USA	None	4	402,214	0.5	201,107	
Bobcat	NC, USA	None	4	921	14	12,889	
Red/gray fox	NC, USA	None	Year round	1,977	4.5	8,895	
E. fox squirrel	NC, USA	None	3	2,931	1	2,931	
Eurasian hare	BW, Germany	None	3	6,422	3	19,266	
Red fox	BW, Germany	None	8	52,836	11	581,196	
Notes.

1 NCWRC.

2 Berichte der Wildforschungsstelle (2020).

* 2018 was a particularly bad hunting year for boar in BW. In 2017, a total of 78,628 individuals were hunted.

Materials & Methods

Study sites

In each country, we chose sites that were as similar as possible in terms of climate, landcover and animal community, however there were key differences between the sites. Specifically, the two countries differ in landcover, landscape structure, climate, mammal community and yard structure. Below, we detail these differences and describe how they might influence our results and if and how we controlled for them in our analysis.

Landcover

In Germany, we sampled sites around the city of Konstanz (pop 84,911), BW. Our study covered an approximate area of 60,000 km2 surrounding the city (Fig. 1) where the landscape was 25.9% forested, 16.8% urban and 30.7% agricultural landcover with an average population density of 259 people/km2. In the United States we focused on a similar sized area (50,000 km2) from Raleigh, NC (pop 464,485) to the east (Fig. 1), that was 41.4% forested, 9.1% urban and 29% agricultural landcover with an average population density of 103 people/km2. Thus, Germany tended to have more people, and NC tended to have more forest. To account for differences in the amount of forest, urbanization and human population between the two sites, we used predictors for the percent urban and percent forested landcover in a 1 km radius (Jung et al., 2020) and their interaction in our relative abundance and occupancy models.

Figure 1 Camera trap locations set within and around two cities: Raleigh, NC, USA and Konstanz, BW, Germany.

We sampled 242 sites in NC and 233 in BW, stratified by urbanized habitat and forest fragments, residential yards and open areas. Cameras ran for 3-weeks, placed in Germany between 2018–2020 and the USA between 2013 and 2019. Portions of this document include intellectual property of Esri and its licensors and are used under license. Copyright ©2021 Esri and its licensors. All rights reserved.

Landscape structure

Both areas had similar levels of gross primary productivity (13083 kg C/square meter BW, 13418 NC in 2015; Hobi et al., 2017) with rolling hills (BW mean elevation = 136 m, NC = 146 m) of mixed deciduous and coniferous forests with similar levels of fragmentation. However, the German landscape featured small, densely settled villages while the American landscape had one larger city with more dispersed housing across rural areas. To account for these differences in the pattern of urban areas across the landscape between BW and NC, we added covariates to our relative abundance and occupancy models representing the size (km2) of the closest urban area and the distance (km) to that urban area.

Climate

The climates of the two sites were similar (BW = coastal, NC = humid subtropical; Kottek et al., 2006) with similar mean annual precipitation (1195 mm BW, 1218 mm NC; Fick & Hijmans, 2017) but with higher mean annual temperatures in NC (7.5C BW, 15.6C NC; Fick & Hijmans, 2017). While we did not directly account for the difference in temperature between NC and BW in our modeling, temperature has never been documented to directly affect large mammal behavior. Instead, we would expect higher temperatures to result in higher productivity, however we found NPP to be similar between the countries suggesting little potential for this difference to affect our results.

Mammal community

Our study focused on the big game species which are both largest and most heavily managed (i.e., bag and season limits) and/or heavily hunted in each region, hereafter referred to as “hunted” species (Table 1). In BW these are roe deer and wild boar (Sus scrofa; hereafter “boar”), both having long hunting seasons with no bag limits (Table 1). In NC these are white-tailed deer, American black bear (Ursus americanus; hereafter “bear”) and wild turkey (Meleagris gallopavo; hereafter “turkey”), all of which have short hunting seasons (1–3 months) and strict bag limits (Table 1). Though different in size (roe deer are smaller), roe deer and white-tailed deer are ecologically similar with similar diets (Vangilder, Torgerson & Porath, 1982; Tixier & Duncan, 1996), habitat preferences (Williamson & Hirth, 1985; Tufto, Andersen & Linnell, 1996) and ability to live close to humans (Etter et al., 2002; Wevers et al., 2020). Deer competitors are absent from NC but present in BW (European fallow deer (Dama dama) and sika deer (Cervus nippon)). However, these competing species are uncommon and therefore unlikely to broadly compete with roe deer and influence our comparisons (Burbaitee & Csányi, 2009). Additionally, large carnivores capable of preying upon deer, especially fawns, are absent from BW but present in NC (coyote (Canis latrans), bobcat (Lynx rufus) and bear; Boone, 2019). However, predation rates on white-tailed deer by these species are relatively low compared to mortality from hunting (Webb, Hewitt & Hellickson, 2007) and do not influence abundances over large scales (Bragina et al., 2019) thus are unlikely to affect our results. Indeed, for both countries deer have little competition or predation, with humans remaining the main regulating factor (McShea, Underwood & Rappole, 1997; Burbaitee & Csányi, 2009).

Yard structure

While we sampled the same number of yards in both countries, we note that German and American yards differ in their size, fencing and vegetation. German yards tend to be smaller than American yards, fenced and highly manicured. American yards tend to be larger and unfenced with more tree cover and natural brush which may support more wildlife. Based on this, we might expect relative abundance indices to be higher in yards in NC over all species, but this should not bias our results relative to hunting since our comparisons between NC and BW for hunted species were restricted to forest fragments.

Field data collection

In NC, we sampled lands owned by NC State Parks, NC Wildlife Resources Commission (Gamelands), US Forest Service and privately owned lands. Owners of private land were 90 private citizens. This study was conducted under NC State Parks permits R12-37 and 2016-0026 and a statewide NC Gamelands permit (no permit number provided by permitting agency) with no field permits required for US Forest Service lands or privately owned lands, just written or verbal permission from the landowner or manager. In Germany, all cameras were run on privately owned lands or lands owned by small conservancies requiring no permits, just written or verbal permission of the landowner or manager. Owners of private land were 212 private citizens. We used a consistent camera trapping protocol between sites (BW and NC) to facilitate comparisons. For each site, trained citizen science volunteers (see Parsons et al., 2018) for details) or staff deployed unbaited camera traps across each study region (Fig. 1). We sampled 242 sites in NC and 233 in BW, with camera placement stratified between hunted and unhunted areas as well as residential yards, forest fragments and agricultural fields (>0.02 km2; Table S1). Information on whether a site allowed hunting came directly from the property owner. In Germany, all hunted areas were forests with no samples from hunted yards or open areas, while in NC some forests, fields and rural yards were hunted (Table S1). We used Reconyx (RC55, PC800, and PC900; Reconyx, Inc. Holmen, WI, USA) and Bushnell (Trophy Cam HD; Bushnell Outdoor Products, Overland Park, KS, USA) camera traps attached to trees at approximately 40 cm above the ground. Trigger sensitivity was set to high for all cameras and we verified that both brands of camera had similar trigger speeds (<0.5 s). Cameras were left undisturbed for 3–4 weeks and then moved to a new location (at least 200 m apart), with sampling taking place over several overlapping seasons and years (2018–2020 Germany, 2013–2019 NC). Cameras recorded multiple photographs per trigger, re-triggering immediately if the animal was still in view. We grouped consecutive photos into one sequence if they were <60 s apart (Parsons et al., 2016), and used these sequences as independent records, counting detections by sequence, not individual photos. Initial species identifications were made by volunteers or staff using customized software (eMammal.org) and all were subsequently reviewed for accuracy before being archived at the Smithsonian Digital Repository. Detection rates for each species at each camera site were calculated as the count/days camera ran, considering groups as a single detection.

Relative abundance

We used a generalized linear regression with a log link, offset for how many days each camera ran, and term for extraPoisson variation, to assess predictors of species detection rates as a measure of relative abundance. We assessed relative abundance for both hunted and unhunted species for which we had >100 detections (n = 10 BW, 9 NC; Table S2). We modeled variation in counts using six covariates (Table S3). To account for differences in the amount of forest and human population between the two sites, we used predictors for the percent urban and percent forested landcover in a 1 km radius (Jung et al., 2020), and their interaction. To account for differences in the pattern of urban areas across the landscape between BW and NC, we added covariates representing the size (km2) of the closest urban area and the distance (km) to that urban area. We used 0/1 indicators for whether a site was a residential yard and whether a site was hunted, respectively.

We fit models in JAGS (Plummer, 2003) via rjags (Plummer, 2016) in R (v3.6.1; R Development Core Team, 2008). We based inference on posterior samples generated from three Markov chains, using trace plots to determine adequate burn-in. All models converged (Gelman et al., 2014) by running for 50,000 iterations following 3,000 iterations of burn-in, thinning every 10 iterations.

Inferring fear

Although experimental manipulation provides the strongest evidence for fear-based responses, many past studies have inferred fear from observational data (e.g., Wooster et al., 2021). Fear response can manifest in many ways, including increased vigilance and avoidance of high risk areas and/or high risk times (Palmer et al., 2017). Here, we inferred “fear” by using a multispecies occupancy model with continuous-time detection process (Kellner et al., 2021) to assess the extent to which wildlife species were using human-dominated habitats and co-occuring with humans, spatially and temporally, while accounting for imperfect detection. We modeled variation in occupancy for each species for which we had at least 100 detections (Table S2) using the same six covariates used for our relative abundance models (Table S3). We modeled detection intensity using two covariates (Table S3): the time latency from a human detection to the next detection of the target wildlife species, used to measure how wildlife responded temporally to humans, and a 0/1 indicator of whether hunting was allowed at the site. We diagnosed correlations in covariates using a Pearson correlation matrix ensuring correlation <0.60. All covariates were centered and scaled prior to analysis. We fit models in R by minimizing the negative log-likelihood using “optim” (R Development Core Team, 2008), with a log-likelihood function implemented in C++ (see Kellner et al., 2021 for model code).

Potential ecological impacts

If wildlife species respond to hunting pressure in terms of changes in abundance, then we would expect concomitant changes in herbivory and/or predation rates, depending on the species being hunted, unless other less hunted species with the same diets are increasing in abundance. Since we were unable to directly measure rates of herbivory and predation, we assessed differences in the relative potential for predation and herbivory based on relative abundances of herbivorous and predatory species and their energetic needs (i.e., body sizes). To assess the relative potential for ecological impacts of each species s we first adjusted the relative abundance to account for larger species being detected over a larger area (Rowcliffe et al., 2011). To relate this to ecological impact, we multiplied by the amount of time spent in front of the camera and the number of animals present, in the case of animal groups. (1) dsj=nsjDj∗tsj∗gsjAsj

where dsj isthe scaled activity of species s on camera j, nsj is the total count of species s on camera j, Dj is the total number of days camera j ran, and Asj is the estimated detection area of camera j, given the body size of species s, following the estimation procedure of Rowcliffe et al. (2011). tsj is the average amount of time species s spent in front of camera j in seconds and gsj is the average group size of species s on camera j.

We calculated the relative ecological impact of each species, specific to three trophic levels (plants, invertebrates, vertebrates), by accounting for their metabolically active mass and diet (Table S4) with the ecological impact of species s on trophic level v given by: (2) Isv=Ms∗psv∗ds ¯

where Ms is the average amount of metabolically active tissue in species s, psv is the percent of the diet of species s made up of items from trophic level v and ds is the average scaled species activity (dsj) for species s from Eq. (1).

Results

Over 7,469 and 5,221 trap nights in BW and NC, we detected mammals and terrestrial birds >80 g a total of 640 and 704 times representing 16 and 20 species, respectively. Hunted species were relatively less abundant with lower occupancy in BW compared to the NC, consistent with the reported 4x greater intensity of harvest in 2018 for roe deer in BW compared to white-tailed deer in NC (Table 1, Fig. S1).

Spatial risk allocation

Both sites had a suite of species, hunted and unhunted, that were detected at high levels of urbanization and near human dwellings (Figs. 2, 3, Table S5). Most species in both countries showed no significant spatial relationship with humans at the site-level except gray fox (Urocyon cinereoargenteus) and eastern cottontail (Sylvilagus floridanus) in NC and red fox and Eurasian badger (Meles meles) in BW which were more likely to use the same sites as humans (Fig. 4, Table S2). Coyotes in NC were less likely to use the same sites as humans, but only in larger urban areas (Table S2).

Figure 2 Relative abundance (detection rate: count/day) for mammal species detected on cameras run in Germany and the USA compared between two levels of urbanization and two habitat types.

Urbanization levels considered were: low (<40% urbanized in a 1km radius) and high (>40% urbanized in a 1 km radius). Habitat types considered were: residential yards and not yards (i.e., forest fragments, open areas). Data are taken from 242 sites in NC and 233 in BW. An asterisk (*) denotes heavily hunted species. Bars show standard error. Hunted species were relatively less abundant at high urbanization but the difference was much greater for German species. Relative abundance for hunted species in the USA were similar between habitat types, while relative abundance for hunted species in Germany was generally lower in residential yards.

Figure 3 Relative potential for ecological impact based on relative abundance, body mass and diet for species captured on camera traps in Germany and the USA.

Herbivores are colored in shades of green, carnivores in pinks and omnivores in blues. We noted an order of magnitude difference in herbivory in yards and open areas in Germany, but similar rates in the USA and in German forests. Potential rates of herbivory were higher in the USA than in Germany for all habitats. Potential predation rates on invertebrates were similar between the countries, being highest in forests in the USA and lowest in forests in Germany. Potential predation rates on vertebrates were higher in Germany, especially in yards, due predominantly to red foxes. Hunting in forested areas reduced the potential ecological impact of mammals across diet types in Germany but increased or did not substantially change it in the USA.

Figure 4 Infographic showing main spatial and temporal relationships with humans for four heavily hunted species.

The position of each species along the color bar indicates the degree of avoidance (red, upper) or attraction (green, lower) to people and their infrastructure. Shown left to right on the “Humans at a site” bar are roe deer and boar in Germany and white-tailed deer and black bear in the USA. Data are taken from camera traps, with 242 sites sampled in NC and 233 in BW. Detailed model results in Tables S2 and S5. Silhouettes: WTD: https://freesvg.org/deer-silhouette-1573809884; Roe: https://freesvg.org/buck; Bear: https://freesvg.org/black-bear-vector-silhouette; Boar: https://freesvg.org/wild-boar-silhouette; Person: https://freesvg.org/vector-graphics-of-simple-family-members-icons; Yard: https://www.clipartmax.com/middle/m2i8H7H7b1N4b1i8_big-image-simple-house-line-art/; House: https://freesvg.org/vector-graphics-of-silhouette-of-a-simple-house; Trees: https://freesvg.org/simple-pine-trees; German flag: https://www.publicdomainpictures.net/en/view-image.php?image=241621picture=german-flag; USA flag: https://commons.wikimedia.org/wiki/File:Possible_52-star_U.S._flag.svg.

Hunted species were relatively less abundant at high levels of urbanization for both sites, with the difference being greater in BW (Fig. 2). Relative abundance for hunted species in NC was similar between habitat types (yard, forest, open), while hunted species in BW were much less likely to be detected in yards than unhunted species (Fig. 3; Table S6). This result was mirrored in our occupancy analyses which showed negative relationships with most hunted species in yards and urban areas, especially for BW (Fig. 4, Table S5).

Temporal risk allocation

Despite few species showing any spatial relationship with humans at the site-level, most species (80% (n = 8) in BW, 56% (n = 5) in NC) showed temporal avoidance of humans (Table S2). Hunted species often showed more temporal avoidance of humans in areas where they were hunted, where most other species temporally avoided humans regardless of hunting (Appendices 2, 7). White-tailed deer showed evidence of temporal avoidance of humans in hunted areas but not in unhunted areas while roe deer showed the opposite pattern (Fig. 4, Fig. S2). Bears showed evidence of temporal attraction to humans in unhunted areas, but not hunted areas (Fig. 4, Fig. S2). Turkeys showed evidence of temporal avoidance of humans, but predominantly in unhunted areas (Fig. 4, Fig. S2). Boars temporally avoided humans in both hunted and unhunted areas, but slightly more in hunted areas (Fig. 4, Fig. S2).

Potential ecological impacts

Due to the high relative abundance of white-tailed deer in NC, potential rates of herbivory were much higher compared to BW. Most striking were the several orders of magnitude lower herbivory rates in yards than forests or open areas in BW due to a lack of roe deer in yards (Fig. 3). Potential predation rates were higher in BW, especially in yards, due to high red fox (Vulpes vulpes) relative abundance while rates in NC were lower and similar across habitats (Fig. 3). Potential predation rates on invertebrates were similar between the countries, being highest in forests in NC and lowest in forests in BW (Fig. 3). Potential ecological impacts in BW across all diet types were much lower in forests that were hunted, while in NC hunted areas had similar or higher potential ecological impacts compared to unhunted areas (Fig. 3).

Discussion

While it seems obvious that increased hunting pressure would affect how animals respond to humans on the landscape, ours is the first study to quantify this by directly comparing the effects of two different wildlife management schemes across a range of human disturbance. Although most of the mammal species are different between the sites, their range of ecological roles are analogous, and the two sites are similar in climate, topography, and land cover. We found several lines of support for the prediction that the more intensive, long-lasting hunting system of Germany contributes to lower relative abundance and differences in risk allocation of hunted species, particularly deer. Although the relative abundance of roe deer in BW appeared to be lowered by intensive hunting, other non-hunted herbivores did not compensate by increasing relative abundance, resulting in lower potential ecological impacts in terms of herbivory in BW hunted areas but not in NC, where hunting does not appear as effective at reducing the relative abundance of white-tailed deer.

Relative abundance

Our prediction that the higher hunting intensity of BW would result in hunted species being relatively less abundant than in the USA was supported. Hunted species had substantially higher occupancy and relative abundance in NC than in BW, suggesting that a more intensive hunting regime may reduce the relative abundance of hunted species and restrict spatial distributions. This also suggests that the presence of deer predators in the NC system did not bias our relative abundance results, consistent with Bragina et al. (2019). The high relative abundance of deer in NC is typical of the eastern portion of the USA where adult deer face little population control from natural predators (Bragina et al., 2019).

Risk allocation

All hunted species showed evidence of spatial avoidance of human modifications to the landscape (i.e., urbanization, yards), with no such avoidance for unhunted species. Our prediction that hunted German wildlife would show more spatial avoidance of humans than hunted American wildlife was supported, with hunted species in BW being relatively less abundant in yards and urban areas that species in NC. This result is consistent with the risk allocation hypothesis which predicts more spatial avoidance in response to threats of higher intensity and longer duration (i.e., year-round hunting vs. 3-month duration), especially in landscapes with smaller, scattered urban areas that can be easily avoided, as we find in BW. The wide suburban sprawl of NC may necessitate a higher level of habituation for hunted species to navigate the landscape and that, along with a lower hunting intensity, shorter duration and common hunting prohibitions in cities and towns, may allow wildlife to maintain activities at a site while avoiding threats temporally. This prediction was supported by our temporal analysis where hunted species in NC showed little spatial avoidance of humans, but more temporal avoidance of humans in hunted areas, especially for the most heavily hunted species: white-tailed deer. However, in BW, heavily hunted roe deer and wild boar showed no temporal avoidance of humans in hunted areas. These results, taken with the spatial avoidance of humans displayed by roe deer, suggest that they are selecting sites with few humans. Indeed, detection rates of humans in forests in BW were low (BW = mean 0.09 people/day, NC = mean 0.21 people/day), making temporal avoidance less necessary. Further study of the fine-scale spatiotemporal dynamics of humans and deer in both countries will help improve our understanding of fear-based responses of wildlife to consumptive recreation.

Potential ecological impacts

Comparing just hunted and unhunted forests showed stark differences in the potential ecological implications of the two wildlife management systems due to difference in relative abundance. German hunted forests had lower mammal relative abundances and thus lower potential predation and herbivory rates. However, hunting in American forests was associated with only marginal declines in expected herbivory, consistent with past studies (Kays et al., 2016). Deer browsing in both countries can be high and has profound effects on forest health and regeneration (Stromayer & Warren, 1997). In Germany, managers often take a “trees before animals” approach that promotes deer hunting as a means of enhancing tree growth (Rooney, 2001). This approach can successfully foster forest regeneration (e.g., Schmit, Matthews & Brolis, 2020) but is dependent on how successfully deer populations can be controlled. Our results suggest that the hunting system of BW is better suited to fostering forest regeneration than in NC where hunting was not associated with a strong reduction in herbivore relative abundance.

In American forests, hunting was associated with increases in predator relative abundance. Given that NC hunters killed over 100,000 predators in 2018 (Table 1), this finding of higher predator relative abundance in hunted forests in NC is non-intuitive. However, light levels of hunting have been shown to increase local predator abundance through increased immigration rates (Gese, 2005) and the potential for increased reproductive output supported by scavenging of carcasses (Mateo-Tomás et al., 2015). This suggests potential indirect community-level effects of hunting through altered social systems and/or productivity of non-target species.

There was a significant difference in the relative abundance of red foxes, one of only two species to occur at both sites. Red foxes are less common in NC and must contend with a variety of competitors (i.e., raccoon (Procyon lotor), gray fox, bobcat, coyote), unlike BW where competitors are rare. The lack of competition, an innate ability to exploit urban habitats (Bateman & Fleming, 2012) and decades of successful rabies vaccination schemes in BW (Storch, Woitke & Krieger, 2005) may benefit red fox populations. We found lower prey relative abundance in BW compared to NC yards which could be a result of higher red fox relative abundances and/or differences in the amount of food and cover present in German yards compared to American yards. Indeed, German and American yards differ substantially in their size, fencing and vegetation, with German yards tending to be smaller, fenced and highly manicured where American yards tend to be larger and unfenced with more tree cover and natural brush which may support small mammal populations.

Study limitations

Our study has some limitations on the interpretation of whether the changes in behavior and relative abundance of hunted animals we observed were caused by the differences in hunting regimes, or by other differences between the countries. The two countries differ not only in hunting style but also in landscape, human population density and yard structure which, although we took into account as much as possible, could nevertheless have affected our results. The ecology of each species could also have influenced our results, including the selection of habitats based on forage quality, presence of conspecifics or population demographic factors (e.g., age structure, density-dependence). While we were unable to account for these factors in the present study, we suggest that further research into their effect on fear-based responses is warranted. Finally, population-specific adaptation should be considered when extending our results to other areas. For example, boars in urban Berlin, where it is difficult or imposible to hunt them, use more urban landscape than in our study area (Stillfried et al., 2017) which could lead to different fear-based responses to humans. This highlights the need for broader study of wildlife and hunting systems to improve our understanding of how hunting practices and human disturbance interact to affect the distribution, abundance and behavior of wildlife populations.

Conclusions

Our results suggest that the more intensive hunting system typical of Germany is associated with lower relative abundance but that the duration of hunting and spatial pattern of humans on the landscape was associated with different fear responses to humans compared to the USA. We noted more spatial avoidance of humans and human structures on the landscape in BW that in NC which should reduce the potential for human-wildlife interactions in an increasingly urban landscape. We found no evidence that unhunted species increase activity or abundance to compensate for declines in their hunted competitors, resulting in ecological benefits in terms of less damage due to herbivory with potential benefits to forest regeneration. Our results show that hunting is a tool that can help reduce potential ecological and social impacts (e.g., wildlife-vehicle collisions, overbrowsing) by changing wildlife abundance and behavior, especially in and around urban areas, and suggests that increasing the intensity of hunting pressure results in more fear of humans. Striking a balance between hunting regimes that effectively regulate wildlife populations and the public’s willingness to tolerate and participate in hunting activities will be important to wildlife management as the world continues to urbanize.

Supplemental Information

Supplemental Information 1 Relative abundance (detection rate (count/day); A) and marginal occupancy (B) for mammal species detected on cameras run in Germany and North Carolina, USA. Data are taken from camera traps (242 sites in USA and 233 in Germany)

Those species that are heavily managed and hunted are shown in blue, bars show standard error. Hunted species were relatively more abundant with higher occupancy in North Carolina, driven by white-tailed deer. Hunted species (i.e., roe deer) were relatively less abundant with lower occupancy compared to unhunted species (i.e., red fox) in Germany.

Click here for additional data file.

Supplemental Information 2 Temporal interactions of four hunted wildlife species with humans measured as the effect of days since the last human detection on wildlife detection intensity compared between hunted and unhunted areas

Data are taken from camera traps run in Germany (European roe deer) and North Carolina, USA (white-tailed deer, wild turkey, bear). Those species shown are the only hunted species to have a significant difference in the relationship between time since last human detection and detection intensity between hunted and unhunted areas. White-tailed deer showed evidence of temporal avoidance of humans in hunted areas only while roe deer showed evidence of temporal avoidance of humans in unhunted areas only. Bears showed evidence of temporal attraction to humans in unhunted areas only. Turkeys showed evidence of temporal avoidance of humans in unhunted areas only. Wild boars showed evidence of avoidance of humans in both hunted and unhunted areas, with higher levels of avoidance in hunted areas.

Click here for additional data file.

Supplemental Information 3 Number of samples (camera trap sites) and average human detection rate (count/day) within three different habitat types (residential yards, open areas and forested areas) between two countries, Germany and the USA

Values are given as Germany—USA and values in parentheses show standard error.

Click here for additional data file.

Supplemental Information 4 Results of a multispecies continuous-time occupancy model examining simultaneous co-occurrence (spatial relationship) and co-detection (temporal relationship) between wildlife species and humans in two countries, Germany and North C

Data are taken from camera traps run in each country (242 sites in NC and 233 in BW), stratified along an urbanization gradient and among yards, forest fragments and open areas. Species marked with a * are heavily managed and hunted. A temporal relationship that depends on hunting (i.e., a significant interaction effect between time since last human detection and whether an area was hunted) indicates that the temporal relationship with humans is different in hunted and unhunted areas.

Click here for additional data file.

Supplemental Information 5 Covariates used in the occupancy and detection rate analyses

For occupancy, each covariate is categorized by whether it was used in the spatial model, temporal model or both.

Click here for additional data file.

Supplemental Information 6 Resources for calculating ecological impacts

We calculated the relative ecological impact of each species, specific to three trophic levels (plants, invertebrates, vertebrates), by accounting for their metabolically active mass and diet. Below are specific reference we used as source material to parameterize this analysis.

Click here for additional data file.

Supplemental Information 7 Coefficients associated with predictors of detection rate and marginal occupancy for species detected in Germany and the United States

Data are taken from camera traps run in each country (sampled 242 sites in NC and 233 in BW), stratified along an urbanization gradient and among yards, forest fragments and open areas. Species marked with a * are heavily managed and hunted. Coefficients marked with a R and O were statistically different from 0 in detection rate and occupancy models, respectively.

Click here for additional data file.

Supplemental Information 8 Mean detection rates (count/day) for each species in each country compared between hunted and unhunted sites and forests, open areas and residential yards. Data are taken from camera traps run in Germany and North Carolina, USA

Parentheses show SE.

Click here for additional data file.

We thank Gregor Wolf for help in the field and in identifying species in Germany. We are indepted to all citizen science volunteers for their support of this project.

Additional Information and Declarations

Competing Interests

Author Contributions

Field Study Permissions

Data Availability

The authors declare there are no competing interests.

Arielle Waldstein Parsons conceived and designed the experiments, performed the experiments, analyzed the data, prepared figures and/or tables, authored or reviewed drafts of the article, and approved the final draft.

Martin Wikelski conceived and designed the experiments, authored or reviewed drafts of the article, and approved the final draft.

Brigitta Keeves von Wolff performed the experiments, authored or reviewed drafts of the article, and approved the final draft.

Jan Dodel performed the experiments, authored or reviewed drafts of the article, and approved the final draft.

Roland Kays conceived and designed the experiments, authored or reviewed drafts of the article, and approved the final draft.

The following information was supplied relating to field study approvals (i.e., approving body and any reference numbers):

This study was performed on private lands in Germany and a mix of private and public lands in NC. Therefore, some NC lands required a permit, which is available for viewing.

The following information was supplied regarding data availability:

The dataset is available at Dryad: https://datadryad.org/stash/dataset/doi:10.5061/dryad.np5hqbzwn.

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
