# Peer review of "Intensive hunting changes human-wildlife relationships"

_PeerJ, doi:10.7717/peerj.14159_

## Round 0.1 · original submission · Major Revisions

· Academic Editor

Major Revisions

Both referees agree that your paper deserves to be published, contributing to the debate on hunting regulations versus forest regeneration. Take into account the suggestions of each one, especially those of Rev 1. We hope that you can soon send us the corrected version so that we can re-evaluate your publication.

·

Basic reporting

The language is mostly clear--I have indicated in the attached .pdf version of the manuscript a few phrases or terms that require clarification.
Literature references are complete and thorough.
The article structure is correct and supplementary data are available. I suggest including some of the occupancy results in the text and figures to support your case and complement the relative abundance results that you do present.

Experimental design

The scope is appropriate, the research questions are relevant.

Validity of the findings

You have an interesting pair of extensive datasets.
I am not qualified to review the statistical analyses you have conducted.
However, my concern is that you have too many differences between the two locations to make meaningful and statistically significant comparisons. You list differences at several points in the manuscript, including in the "study limitations" section.
Therefore it may be more compelling to focus your analysis on comparing the two deer species, rather than adding in the broader suite of species and adding in potential ecological impacts. If you wish to keep these other pieces, then more justification is required and clearer definitions of terms and more details of the analyses you conduct.

Additional comments

I have made additional comments on the attached .pdf version of your manuscript, for your consideration.

Reviewer 2 ·

Basic reporting

Well written and clear professional writing, literature is sufficient (one needs updating), data shared, code was not but is linked to another publication - self-contained and relevant hypotheses

Experimental design

Meets all criteria and uses state of art models

Validity of the findings

Useful and potentially very important findings that can aid in the debate of hunting regulations versus forest regeneration

Annotated reviews are not available for download in order to protect the identity of reviewers who chose to remain anonymous.

---

## Round 0.2 · accepted · Accept

· Academic Editor

Accept

The authors have completed all the suggestions made by the reviewers, I am happy to accept the paper

·

Basic reporting

The authors in their revised manuscript have addressed all of my concerns.

Experimental design

The authors in their revised manuscript have addressed all of my concerns.

Validity of the findings

The authors in their revised manuscript have addressed all of my concerns.

Additional comments

Thank you for your quick and very detailed response to my comments on your first manuscript. In your revised version here you have addressed all of my concerns.
Please just complete a copy edit to catch any typographical errors.

Reviewer 2 ·

Basic reporting

The language is professional and clear, the literature cited is sufficient and gives a clear summary of the background. The article is well structured and has a self-contained hypothesis.

Experimental design

The manuscript fits well within the scope of the journal with a state-of-the-art model to answer gaps in knowledge on spatiotemporal interactions from camera trap data, that previously could only be investigated in two unassociated models.

Validity of the findings

The findings are new and gives good insight into the differences in pressure from hunting and landscape use in the US and Germany.

Annotated reviews are not available for download in order to protect the identity of reviewers who chose to remain anonymous.